# *Brassica napus* Roots Use Different Strategies to Respond to Warm Temperatures

**DOI:** 10.3390/ijms24021143

**Published:** 2023-01-06

**Authors:** Marta Boter, Jenifer Pozas, Jose A. Jarillo, Manuel Piñeiro, Mónica Pernas

**Affiliations:** Centro de Biotecnología y Genómica de Plantas (CBGP), Universidad Politécnica de Madrid (UPM) and Instituto Nacional de Investigación y Tecnología Agraria y Alimentaria-Consejo Superior de Investigaciones Científicas (INIA-CSIC), Campus de Montegancedo, 28223 Madrid, Spain

**Keywords:** *Brassica napus*, climate change, crop adaptation, heat-shock response, root traits, temperature, comparative transcriptomic analysis

## Abstract

Elevated growth temperatures are negatively affecting crop productivity by increasing yield losses. The modulation of root traits associated with improved response to rising temperatures is a promising approach to generate new varieties better suited to face the environmental constraints caused by climate change. In this study, we identified several *Brassica napus* root traits altered in response to warm ambient temperatures. Different combinations of changes in specific root traits result in an extended and deeper root system. This overall root growth expansion facilitates root response by maximizing root–soil surface interaction and increasing roots’ ability to explore extended soil areas. We associated these traits with coordinated cellular events, including changes in cell division and elongation rates that drive root growth increases triggered by warm temperatures. Comparative transcriptomic analysis revealed the main genetic determinants of these root system architecture (RSA) changes and uncovered the necessity of a tight regulation of the heat-shock stress response to adjusting root growth to warm temperatures. Our work provides a phenotypic, cellular, and genetic framework of root response to warming temperatures that will help to harness root response mechanisms for crop yield improvement under the future climatic scenario.

## 1. Introduction

The effects of climate change are threatening crop productivity across the globe. The increasing incidences of heat waves, drought, and other extreme weather events experienced worldwide are negatively affecting agricultural production [1,2]. Feeding the world’s population will require a significant rise in food production against the backdrop of these climatic constraints [3,4]. In view of a future increase in food insecurity, agriculture needs to find new ways to adapt crops to adverse environmental changes [5,6,7]. One of the major alterations triggered by climate change is a global trend of warmer temperatures [6,8]. Elevated ambient temperatures have profound effects on plant physiology and development, leading to substantial declines in crop yield and quality [9,10]. Although crops are heterogeneously affected, higher temperatures generally shorten crop growth periods, affect photosynthetic rates, reduce plant shoots and root biomass, promote fruit senescence, decrease seed numbers and sizes, and alter seed composition [11,12,13,14]. Even though the predicted increase of a few degrees in ambient temperature can have profound effects on crop growth and yield, information on how crops adapt to warmer temperatures is still scarce [15].

Roots are the main organs that control nutrient and water uptake, and changes in soil temperatures alter these processes, limiting crop growth. Root systems are also highly plastic in response to environmental conditions and can modulate different physiological and morphological traits to adapt their architecture and functionality to disadvantageous environments [16,17,18]. Although root response to elevated temperatures could vary between crops, in general, it causes a decrease in the primary root length and an alteration of lateral roots’ growth, number, and angle of emergence, reducing the surface between root and soil [19]. This reduction in root growth is usually accompanied by a lower root-to-shoot ratio and a reduction in root carbon allocation. As a consequence, the nutrient and water uptake conducted by the roots is compromised, and crop yield is severely affected [20]. Given the difficulty in characterizing root system architecture (RSA) in the field environment, root traits associated with improved responses to rising temperatures have been classically underrepresented in breeding programs. Additional complexity has come from the soil temperature dynamics faced by crops in the field. Soil temperature varies seasonally and daily, mainly due to variations in air temperature and solar radiation. These changes particularly alter plant metabolism and growth, affecting root development and nutrient uptake, but also fungal pathogen patterns of colonization in the field, impairing crop yield [21,22,23]. Increasing data on roots’ contribution to crop yield, together with the potential to exploit their plasticity for crop adaptation, have driven a rise in the use of below-ground traits to obtain more suitable crops [24,25,26,27,28,29].

One of the crops more severely affected by environmental factors, particularly rising temperatures, is oilseed rape (*Brassica napus*). Oilseed rape is the second most important oilseed vegetable after soybean, used both as an oil and a protein crop [30]. Reductions in yield of up to 40% have been already reported due to elevated temperatures [31,32]. In the field, higher-than-optimum temperatures determine the duration of the growth stages and negatively affect seedling emergence and reproductive stages, which ultimately leads to a reduction in productivity [33,34]. Understanding how increasing temperatures affect root traits would help in designing better strategies for improved production in the coming decades. However, to successfully incorporate root traits in the generation of new cultivars, a better understanding of the genetic, cellular, and molecular mechanisms regulating RSA response to differences in soil conditions, including changes in temperature, is urgently needed [35]. In particular, we are interested in the changes that root architecture undergoes in response to warmer soils that represent the most likely scenario predicted by a future increase in global temperatures driven by climate change [8]. Oilseed rape’s optimum growth and flowering temperature is considered to be 20–21 °C, whereas 29–30 °C corresponds to warming temperatures [36]. With the aim of identifying beneficial traits of root responses to warm temperatures, we characterized the phenotypic variability associated with RSA modulation at 29 °C in a panel of spring oilseed rape (SOSR) varieties. We analyzed root responses at early stages of plant growth when the crops are more vulnerable to a poor growing environment, and temperatures particularly affect seedling establishment of temperate crops, including OSR [37,38]. Thus, in this study, we defined the most common RSA changes provoked by warm temperatures and captured the differential strategies that oilseed rape roots use to modify their architecture to this environmental condition. *B. napus* roots’ response tends to combine an increase in the extent and depth of their primary roots with changes in the distribution and elongation of their secondary roots. As a result, warming conditions lead to wider and deeper root systems that extend their area of soil exploration, improving their probability of accessing water and nutrients. Using comparative anatomical analysis, we also associated these RSA modifications with changes in the regulation of specific cellular mechanisms. In response to warm temperatures, *B. napus* plants rearrange their root growth by combining increased meristematic cell division with enhanced cell elongation. We also used comparative transcriptomic analysis to define the main genetic pathways triggered during this root response to warm temperatures. Besides the activation of the general mechanisms of high-temperature sensing and signaling, we uncovered a key role of cell growth regulators in the control of warmth-induced root growth. Furthermore, we propose that an efficient attenuation of the initial heat-shock response would be required for root response to warming temperatures in *B. napus* roots. In summary, the information gathered in this study sheds light onto the strategies adopted by oilseed rape roots to confront warming conditions and has helped us to identify putative genetic determinants of root response that might be used to secure yield stability under the challenging environmental conditions driven by climate change.

## 2. Results

### 2.1. Brassica napus Roots Adopt Different RSA Configurations to Increase Their Ability to Explore the Soil in Response to Warm Temperatures

To explore the phenotypic variability associated with RSA modulation by warming temperatures, we monitored the roots of a panel of 10 SOSR varieties grown for 7 days at 21 °C (optimal growth conditions) and 29 °C (warm conditions) on a pouch-and-wick system [39]. We quantified a wide set of root traits to analyze the changes in the extent of exploration, intrinsic size, distribution, and shape of root networks in response to warm temperatures. We observed that in roots grown under warm conditions, the majority of root traits are significantly changed (83.3%) (Table 1 and Appendix A). Traits encompassing all different aspects of RSA organization were altered, but a distinct, significant increase in in extent- and size-related traits in most of the *B. napus* genotypes analyzed suggested that warming conditions led to extended and bigger root systems. Thus, we observed a significant increase in network depth, width, and consequently, convex area, indicating that the exposure of roots to warm temperatures leads to a larger extent of the medium being explored (Figure 1A and Appendix AA). Similarly, we found increased values of network length, area, and surface area, also representing an increase in root biomass and the root surface in contact with the medium (Figure 1A and Appendix AB, Appendix A). Then, we quantified the relative shape of the root network and observed minor changes in aspect ratio and network width-to-depth ratio values (Figure 1A and Appendix AE, Appendix A). Due to the topology of the *B. napus* root system, network depth values mainly represent the elongation of primary roots, whereas network width represents the elongation of secondary roots (Figure 1D). Thus, these results suggest that overall, *B. napus* roots respond to warming temperatures by altering the growth of both primary and secondary roots. Finally, the analysis of root traits associated with network distribution showed a reduction in network length distribution, but not significant changes in network bushiness (Figure 1A and Appendix AC, Appendix A). These results indicate that the increase in root biomass and growth triggered by warm temperatures is not equally distributed along the root system. Hence, the observed decrease in network length distribution evidences a trend to increase the amount of the network accumulated in the top portion of the RSA.

More interestingly, together with a common effect of warm temperatures, we also found that there was a significant effect of genotype on all the evaluated root traits (Table 1). In order to identify and characterize the genetic variability associated with RSA response to warming, we performed multivariate principal components analysis (PCA) and hierarchical clustering (HC). Using all previous trait measurements, we found that 69% of the variability in temperature-dependent alteration of RSA traits among genotypes could be explained by the first two components, 46% and 22, 5% in PC1 and PC2, respectively (Figure 1B). Traits with higher influence in the PC1 were size-related traits, such as network surface and network area, followed by network perimeter and network length. The high contribution of size traits indicates that there is a high variation in these traits between the varieties, and accordingly, values ranged from 0.98 to 1.67 for the network surface or from 0.99 to 1.66 for the network area in Drakkar and Wesway varieties, respectively (Figure 1A,B and Appendix AB and Appendix A). Likewise, shape traits also had an important contribution to PC1, with aspect ratio values from 1.37 to 0.83 in Wesway and Westar (Figure 1A). Interestingly, high values of these two traits correlated with high values of extent traits, such as the minor ellipse axis and network width in PC1, but also with negative high values of the major ellipse area or network depth in the PC2. These data suggest that most varieties change the depth of their root system mainly through the growth of their primary roots (major ellipse area vs. network depth traits), but key differences in the growth of the secondary roots account for the diversity found in the width of the network (minor ellipse area vs. network width) (Figure 1A,B and Appendix AA, Appendix A). Altogether, these results imply that each variety uses different arrangements of their root traits to increase medium exploration. Consistently, when we positioned the genotypes according to their PC score, we identified three root response groups. A first group comprised the Westar, Karat, Dux, and Line varieties that showed low scores in all PCs but particularly in PC2 scores, network width, and minor ellipse area, thus representing a low response with no preferential top allocation of root networks. Although having similar PC1 scores, the Drakkar variety was allocated separately from this group due to the lowest scores of network depth (1.03) and major ellipse area (0.60), together with high network width-to-depth ratio (1.15). Consequently, the growth of secondary roots was increased, but not that of the main roots. Interestingly, Drakkar had high network solidity and secondary root density values, suggesting that although their roots explore less area, plants of this variety augmented their surface of interaction with the medium through an increase in the number of secondary roots (Figure 1A,B and Appendix AA–E, Appendix A). Therefore, Drakkar represents a variety that develops wider but not deeper root networks in response to warm temperatures. A broad third group was constituted by varieties distinguished by similar PC1 and PC2 scores (Industry, Marnoo, Fido, Wesway, and Duplo) that coincide with genotypes showing moderate changes in most of root traits but a very substantial increase in extent traits, major ellipse area, and network depth. Thus, this group mainly represents varieties growing significantly more extensive and deeper roots in response to warming. Accordingly, this group contains the variety Duplo, with the highest score for convex area (2.1), resulting from high network width (1.47) and length (1.84) scores. Therefore, Duplo represents a positive root response to warm temperatures by displaying a root system covering more extension due to a combination of wider and deeper roots (Figure 1A,B and Appendix AA–E, Appendix A). Hierarchical clustering (HC) of the genotypes based on these RSA traits further supported these PCA results. Thus, the varieties were grouped in three clusters according to the prevailing combination of traits displayed to respond to warming (Figure 1C). This clustering underlies the same phenotypic categorization of the varieties obtained in the previous analysis. In summary, the variability in RSA response in *B. napus* roots reflects the plasticity of the root system to respond to different environmental conditions. Thus, *B. napus* roots seem to take advantage of this plasticity by deploying different modifications of root traits to better exploit their surrounding environment.

### 2.2. Combinatory Changes in Cell Elongation and Cell Division Drive Differential Root Response to Warm Temperatures

We have shown that an increase in root growth is a common response to warm conditions in *B. napus*. To identify the cellular processes that are responsible for this growth, we carried out a comparative analysis of the primary root growth process between two *B. napus* varieties, Drakkar and Duplo, that differ significantly in their primary root response to warm temperatures (network depth 1.03 and 1.41, respectively) (Figure 1A,D and Appendix A). First, we monitored the primary root growth dynamics of the two varieties in seedlings grown at 21 °C and 29 °C. Warming temperatures led to a significant increase in primary root length in Duplo, whereas no differences in length were detected in Drakkar after exposure to 29 °C compared to 21 °C (Figure 2A). Moreover, this differential root growth response was also maintained when *B. napus* seedlings were grown in perlite pots and exposed to warm temperatures for longer time periods. After 15 days, the root length ratio between 29 °C and 21 °C was similar to the values we observed at 7 days, 0.81 in Drakkar and 1.36 in Duplo (Figure 2B), suggesting that root growth response triggered by warm temperatures might be maintained through the whole seedling growth.

Root length growth is the result of combined cell proliferation in the meristem and rapid longitudinal cell expansion in the elongation zone. Hence, we measured these cell parameters in Drakkar and Duplo roots after 7 days exposed to warm (29 °C) and control (21 °C) temperature conditions. First, we measured root meristem parameters. Confocal microscopy on mPS-PI-stained roots revealed that both varieties displayed shorter root meristems at 29 °C than at 21 °C, at 776 µm vs. 612 µm in Drakkar and 731 µm vs. 609 µm in Duplo (Figure 3A and Figure 4B). This decreased meristem size was correlated with a reduced number of meristematic cells in both genotypes, 75 vs. 58 and 73 vs. 56 in Drakkar and Duplo, respectively. However, when we measured the average meristematic cell length, we found that it was higher at 29 °C than at 21 °C in Duplo, but it remained similar in Drakkar at both temperatures (Figure 3A). A cell size frequency distribution analysis revealed that in Duplo, warming resulted in a reduction in the number of shorter cells and a consistent increase in the frequencies of longer-sized cells, from 8 (24.08% at 21 °C to 18.3% at 29 °C) to 12µm (16.9% at 21 °C to 22.4% at 29 °C) (Figure 3B). In contrast, Drakkar displayed a similar distribution of cell length frequency (from 22.6% to 22.5% or from 19.7% to 20% for cell length frequencies corresponding to 8 or 12 µm, respectively) at 21 °C and 29 °C. The root meristem is divided into the apical and basal meristem according to the upward position from the root tip. Cells in the apical meristem are continuously dividing and expanding at a constant rate, whereas cells at the basal meristem also divide, but rapidly increase their size to exit the meristem through the transition zone. Since we found differences in average meristem cell length in Duplo, we were interested in mapping where these differences arise in the meristem. We found no significant differences in cell length, cell number, or meristem length in the basal meristem between 21 °C- and 29 °C-treated roots in any of the genotypes (Figure 3C). However, when we recorded the average cell length of the meristematic cells according to their position in the root meristem from the initials close to the quiescent center (QC) up to the meristem transition zone, we found that cells sharply increased their cell length earlier at 29 °C, at 411 µm compared to 561 µm from the QC in Drakkar and at 385 µm compared to 517 µm at 21 °C in Duplo, suggesting that the boundary between the apical and basal meristem starts closer to the QC when roots were grown at warm temperatures (Figure 3D). Consistently, the meristem length and total number of cells of the apical meristem were also reduced in both genotypes (58 vs. 42 in Drakkar and 57 vs. 38 in Duplo), and the average cell length increased significantly only in Duplo (Figure 3C). Altogether, these data indicate that warming temperatures provoke a shortening of the root meristem caused by a reduction in the apical meristem size.

Cell production in the meristem mainly relies on two factors, the number of dividing cells and their rate of cell division. Since the meristem of both genotypes contained less meristematic cells in response to warm temperatures, we analyzed whether there were changes in their rate of cell division that could account for the differential root growth response in both varieties. We quantified the differences in cell division rates in response to warm temperatures by scoring double-labeled-EdU and DAPI nuclei cells using confocal microscopy. First, we monitored warmth-induced changes in DNA replication by measuring EdU incorporation rates that represent the portion of meristematic cells in the S-phase among all meristematic cells (EdU-labeled nuclei/DAPI-labeled nuclei). As shown in Figure 3E,F, EdU incorporation ratios were not significantly different between roots grown at 21 °C and 29 °C in either of the two varieties, indicating that warming temperatures did not strongly affect DNA replication. Then, we measured the relative number of mitotic cells (mitotic index) by scoring M-phase EdU-labeled nuclei (number of EdU-labeled M-phase nuclei/total EdU-labeled nuclei). We found a significant increase in mitotic figures in Duplo roots grown at 29 °C compared to 21 °C, with the EdU-related mitotic index rising from 2.41 at 21 °C to 5.49 at 29 °C, but we did not detect any difference in Drakkar (Figure 3G). Altogether, these results suggest that Duplo meristems compensate for the reduction in meristematic cell numbers in response to warm temperatures by increasing their cell division rate.

Next, we quantified the number and length of the cells in the root elongation zone (EZ), where cells no longer divide but expand rapidly. When we examined the elongation zones of mPS-PI-stained roots grown at 29 °C and compared these with roots grown at 21 °C, we observed a significant decrease of 44% in the length of Drakkar EZ, whereas no change was observed in Duplo (Figure 4A,B). The differences in EZ length correlated with a decrease in the number of elongating cells, from 33 to 24 cells (27%), and a reduction in average cell length, from 92 to 76 µm (18%), in Drakkar. By contrast, in Duplo, a significant increase in average cell length, from 72 to 83 µm (14%), compensates for the decrease in cell number, maintaining the EZ length (Figure 4A,B). Accordingly, we observed a clear increase in the relative frequencies of longer cells in Duplo EZ compared to Drakkar EZ (Figure 4C). The analysis of the distribution of cell size relative to cell position in the EZ of both varieties showed that, as expected, at both temperatures, cell length increased as cells were reaching the differentiation zone. However, significant differences among the genotypes were revealed at 29 °C. Cell length increased more sharply along the root EZ in Duplo than in Drakkar (Figure 3D), suggesting that cells elongated more quickly in Duplo compared to Drakkar EZ in response to warm temperatures. Consequently, the first cell in the differentiation zone was significantly bigger in Duplo at 29 °C than in Drakkar, further evidencing the differences between both genotypes (Figure 4E). These results showed that warm temperatures enhance cell elongation in Duplo but not in Drakkar EZ. Our data suggest that *B. napus* roots might combine different cellular changes, such as increasing cell proliferation in the meristem followed by enhanced cell elongation, to integrate the root growth response to warm temperatures. However, this combinatorial strategy must be differentially applied between varieties, since spatial root growth response varies among genotypes. Thus, when we analyzed the cellular response of Marnoo, another genotype that increased the length of their primary root in response to warming, although to a lesser extent than Duplo (Appendix AA), we found that the meristem length, cell number, and average cell length did not display significant changes when grown at 29 °C (Appendix AB). In contrast, in the EZ, average cell length was significantly increased compared to 21 °C (Appendix AC), suggesting that cell elongation is the main cellular mechanism employed in this variety to respond to warming. These results further support the hypothesis that *B. napus* deploys different cellular changes to respond to adverse temperatures.

### 2.3. Balanced Regulation of Transcriptional Temperature Response Is Crucial to Adjusting Root Growth to Warming Conditions

We have shown that *B. napus* roots modify their developmental program to adjust their growth to warming temperatures. To understand the genetic regulation of this response, we compared the root patterns of gene expression of the *B. napus* varieties Drakkar and Duplo using transcriptomic analysis. Correlation analysis of the differences in gene expression with the differences in primary root elongation observed in these two different genetic backgrounds will allow us to identify the major transcriptomic changes that define the primary root response to warming temperatures, as well as some of the gene regulatory networks that may contribute to the differences in RSA detected in these varieties. With this aim, we carried out RNA sequencing (RNA-seq) of Drakkar and Duplo root tips grown at 21 °C and 29 °C for 7 days. Differential gene expression analysis of these data identified 930 differentially expressed genes (DEGs) in Duplo and nearly three times more genes, 2605, in Drakkar (DEGS, adjusted *p*-val < 0.05 and −1 > log2FC > 1) (Figure 5A,B and Appendix A). Interestingly, 77.7% of the Drakkar DEGs were upregulated, whereas 58.5% were induced in Duplo, suggesting that an active transcriptional reprogramming is triggered in response to warm temperatures. Comparative analysis of these DEGs showed a low correlation between the gene expression responses to warming temperatures of the two varieties (Appendix AA). Only 12.2% of Drakkar DEGs were shared with Duplo DEGs, whereas 34.4% of Duplo DEGs overlapped with Drakkar DEGs (Figure 5C), indicating that there are significant differences in the temperature-dependent transcriptome reprogramming between the two varieties. Hierarchical clustering of all the genes with altered expression at 29 °C in both varieties revealed six major patterns of transcriptional response to warmer temperature (Figure 5B and Appendix A). Cluster 1 (1105 genes) and Cluster 5 (422 genes) contained genes predominantly induced or repressed in both varieties, respectively. Gene ontology enrichment analysis showed that Cluster 1 was enriched in genes related with the response to oxidative stress (*BnPRX71*), fatty acids (*BnACX2*), and sugars (*BnSUC7*), as well as ethylene (ET) biosynthesis (*BnSAM1*, *BnACO1*) and splicing machinery (*BnSR30*), whereas Cluster 5 contained genes related to metabolic processes and beta-glucosidase activity (Figure 5C and Appendix A). These clusters represent the common patterns of gene expression between genotypes, and therefore, either the transcriptional activation (Cluster 1) or repression (Cluster 5) of these groups of genes underlies their shared temperature induced response, as confirmed by qPCR (Figure 5B,C, Figure 6 and Appendix AB,C). Remarkably, the biological processes that were overrepresented, such as oxidative, lipid, and splicing pathways, have been defined as part of the common mechanisms of temperature sensing and signaling in plants [20,40,41].

The remaining clusters comprised genes showing differential expression between the two genotypes and are therefore more likely to contain the genetic determinants of the variability in root response to warming conditions. Clusters 3, 4, and 6 represented genes that had an opposite expression in Duplo compared to Drakkar. We found genes that were repressed in Drakkar but less repressed or induced in Duplo (Cluster 3), induced in Duplo but less induced in Drakkar (Cluster 4), or repressed in Duplo but induced or less repressed in Drakkar (Cluster 6). Regarding the type of genes contained in these clusters, Cluster 3 (260 genes) was enriched in genes related to nitrate assimilation, metabolism, and transport (*BnNRT2.1*) and to cell wall-related processes. Cluster 4 (352 genes) included genes involved in peroxidase activity, cell wall biogenesis and organization (*BnXTR6*, *BnEXPA8* and *BnXYL4*), and abscisic acid (ABA) signaling (*BnPYL4* and *BnPYL5*). Thus, in Duplo, the transcription of several cell growth regulators is activated or repressed to enhance root growth, whereas in Drakkar, this response is not fully accomplished (Figure 5B and Appendix AB). Finally, Cluster 6 (153 genes) encompassed genes related with either light-signaling pathways or negative regulation of photomorphogenesis (*BnPIL2*) (Figure 5C and Appendix AB). Comparison and validation by qPCR of the differential expression patterns of key genes belonging to these biological processes in each cluster confirmed that *B. napus* roots specifically modify their transcriptional programs to adjust their growth to respond to warming temperatures by coordinating the activation of temperature sensing/signaling and cell growth regulatory pathways (Figure 6 and Appendix AB,C).

Lastly, Cluster 2 (922 genes) was the largest differential cluster, representing more than half of the genes (54.66%) with opposing expression patterns. This cluster comprised genes that are highly induced in Drakkar but less induced or repressed in Duplo, and it was enriched in genes related to the response to hydrogen peroxide (*BnCAT3*) and high light intensity (*BnGols1*). However, the most represented GO of this cluster, corresponding to 56 genes, was “response to heat”, and this included several core heat-shock response genes (HSR), such as heat-shock proteins (HSPs) (*BnHSA32*, *BnHSP15.7*), chaperones (*BnP23*) and heat-shock transcription factors (HSFs) (Figure 5C and Appendix A). Moreover, this group of heat-shock response genes not only showed the highest fold changes in Drakkar DEGs (13 HSR genes out of 15 genes with log2FC > 8, 86%) (Figure 7A), but also the highest differences between Drakkar log2FC and Duplo log2FC (13 HSR genes out of 19 genes with ∆DK log2FC/DP log2FC > 8, 68.4%) (Figure 7B). The negative correlation between the high induction of HSR gene expression and the reduction in primary root growth in Drakkar, together with the positive correlation of increased primary root elongation with the repression of the same genes in Duplo, as assayed by qPCR, strongly suggested that a tight control of HSR is required to stimulate warmth-triggered primary root growth (Figure 6A and Appendix AB,C). Consequently, we speculated that when roots are exposed to warming temperatures, an initial activation of HSR is triggered in both varieties, but subsequent HSR repression is initiated in Duplo and not in Drakkar. To test this hypothesis, we analyzed the transcriptional dynamics of several HSR genes belonging to Cluster 2 at 24 h, 48 h, 4 days, and 7 days after being exposed to the control temperature, 21 °C, and 29 °C in both varieties. Thus, we monitored the pattern of expression of four well-known core HSR genes, including the small heat-shock protein, *BnHSP17.6*; the ATP-dependent chaperone of the HSP70 family, *BnHSP70*; the mitochondrion-localized small heat-shock protein, *BnHSP23.6M*; and the HS-induced galactinol synthase, *BnGolS1*, by qPCR (Figure 7C) [42,43]. As expected, we found that all these genes started to increase their expression at 24 h after 29 °C exposure. In Duplo, this increase was constantly maintained until reaching the maximum at 4 days, when the expression levels were reduced (*BnHSP17.6* and *BnHSP70*) or steadily maintained (*BnHSP23.6* and *BnGols1*) up to 7 days. On the contrary, in Drakkar, this gene induction gradually increased from 24 h without declining until day 7 when the genes reached the maximum differential expression levels compared to Duplo (Figure 7C). These gene expression patterns correlated well with the opposite primary root growth response to warm temperature observed for these two varieties and suggest that the phenotypic RSA response could be associated with differences in the transcriptional dynamics of the heat-shock response. Based on these results, we propose that in the roots, warming temperatures lead to an early activation of the HSR. Once this response is triggered, roots attenuate this stress response to avoid its detrimental effect over growth. We postulate that roots of oilseed rape varieties such as Drakkar that could not counterbalance their temperature stress response and maintained high levels of HSR gene expression strive to readjust their root growth. Further analysis of the effect of the sustained expression of some of the key HS regulators identified in this study on root response to warming will contribute to corroborating this hypothesis.

## 3. Discussion

Root system architecture determines the root capacity of supplying water and nutrients to the plant. Under changing environmental conditions, RSA needs to undergo rearrangements to effectively reach water and nutrient-rich patches in the soil. The identification of root traits responsible for this response is crucial for developing climate-adapted cultivars [18,35]. Several effects on root system architecture due to changes in soil temperatures have been previously described in crops. Thus, a reduction in root growth associated with higher temperatures has been shown in maize, sorghum, rice, and potato [44,45,46,47]. On the other hand, changes in adventitious and lateral roots’ initiation and elongation have also been described in potato, maize, sweet potato, and cassava [48,49,50]. However, few studies have focused on characterizing the root response to warm temperatures that is predicted to be the most likely temperature condition confronting crops. Our results suggest that constant warming temperatures enhance some beneficial root traits related to the roots’ ability to explore their environment in several oilseed rape varieties at early stages of seedling development. These root traits included an increase in the extent and depth of primary roots, changes in the distribution and elongation of secondary roots, and an increase in the root surface area due to the combination of primary and secondary root extension. Bigger root systems assist water and nutrient uptake to support active growth during seedling establishment. They also increase the growth of the crop canopy, enhancing seedling survival to pest and environmental stresses and improving their competition with weeds. Consistently, some of the RSA changes that we have described in response to warming are coincident with root traits already associated with QTLs that are found in cultivars that are more tolerant to high temperatures and drought stress in some crops, such as rice, wheat, and barley [51,52,53,54,55]. These QTL-associated traits correspond to higher root length or enhanced superficial root distribution, suggesting that the similar root responses displayed by some of the *B. napus* varieties analyzed could be positive root traits for tolerance. Moreover, in other species, larger root size and fast early root expansion have been linked to adaptive advantages to temperature by enhancing plant competition in the field [23]. Contrary to this observed positive RSA response to warming, the exposure of roots to temperatures that are higher than optimal generally causes a negative effect in the overall root growth, hinting that plants have a narrow temperature margin between triggering an adaptation or stagnation root growth response [20,55]. Harnessing the modulatory mechanism responsible for this root growth readjustment may help to develop crops that are equally adapted to either heat waves or an increase of a few degrees in temperature triggered by climate change [56]. Our analysis of root response variability to warm temperatures in *B. napus* seedlings also uncovered differential growth responses between primary and secondary roots. At warmer temperatures, the elongation of primary and secondary roots was not concertedly enhanced in all varieties, but it underlies Drakkar’s divergent response, consisting of the elongation of secondary roots, though not of primary roots (Figure 1B,C). These phenotypic responses suggest that specific root developmental signals could differentially modulate the growth programs of primary and lateral roots to warming temperatures. Several hormones, such auxins, ABA, brassinosteroids, or gibberellins, have been described as modulators of differential responses to environmental stress displayed by primary and secondary roots in several plant species [41,57,58,59]. Since some regulatory components of these hormonal pathways were differentially expressed in Drakkar primary roots compared to Duplo, one tempting possibility is that a hormonally dependent mechanism could be responsible for their contrasting growth responses to warm temperatures (Figure 5C and Appendix A). Further comparisons of primary and lateral roots’ specific transcriptional programs will be needed to elucidate the regulatory mechanisms enabling their distinct growth response.

On the other hand, the shortening of root meristems upon exposure to warm temperatures is a common response mechanism between varieties. Negative effects of elevated temperatures on root meristem size have been described previously in other species, causing a decrease in root growth [60,61,62]. However, in *B. napus* roots, the reduction in meristem size by warm temperatures positively correlates with an increase in longitudinal root growth. Hence, the alteration of meristem length should be compensated for by changes in either cell division rates or cell length. Coordination between size and cell division to maintain meristem structure has been already described in Arabidopsis shoots [63,64]. Warm temperatures increase the size and number of mitotic root cells, so regardless of the differential organization of both meristems, a similar compensatory mechanism might also maintain root meristem homeostasis in *B. napus* roots. Cell growth and cell cycle need to be coordinated in each cell, so warm temperatures must affect the progression of the cell cycle [65,66]. It has been already described that cell cycle length is dependent on environmental conditions, including increased temperatures [67]. Acute heat stress causes G2 arrest in Arabidopsis, but in maize root cells, the cell cycle phases progress more rapidly due to a shorter cell cycle time when exposed to 30 °C [68,69]. Although our results point to a warming-induced acceleration of the cell cycle, to determine precisely how warm temperatures control cell cycle progression, the application of recently developed methods for analyzing spatial and temporal dynamics of the cell cycle “in vivo” in roots would probably be needed [70,71].

A second cellular change driving *B. napus* root growth in response to warm temperatures is a progressive increase in cell elongation. The remodeling and loosening of cell walls by different cell wall enzymes are essential steps to promoting cell elongation. Accordingly, warm temperatures preferentially induced the expression of several of these enzymes, such as expansins, XTHs, and XYLs in Duplo roots, where cell elongation increased more sharply [72]. Brassinosteroids (BRs) elicit cell expansion by altering cell wall properties under various environmental conditions [73,74]. In particular, BRs have been shown to mediate an increase in root cell length at warm temperatures in Arabidopsis [61]. Coincidentally, *BnSTE1* expression, a BR biosynthetic enzyme, as well as some orthologous groups of BR-regulated cell wall enzymes (*BnXTH24*, *BnEXPA8*, and *BnXTR6*) were induced in response to warming in *B. napus* roots [75]. These results suggest that a BR-regulated pathway might also control warmth-induced cell elongation in this crop. It has also been shown that cytokinins (CKs) negatively control cell elongation in the root elongation zone in Arabidopsis [76]. We have observed that warming induces two negative regulators of CKs, *BnCKX1*, a cytokinin oxidase/dehydrogenase that catalyzes the degradation of CKs, and *BnKMD1*, a member of a family of F-box proteins called KISS ME DEADLY (KMD) that targets type-B ARR proteins for degradation. Moreover, we detected that the expression of *BnHB-3*, a downstream effector of CK signaling, is also differentially repressed in response to warm temperatures, suggesting that changes in cell elongation may be mediated by the downregulation of CK response (Figure 5 and Appendix A).

Regulation of gene expression is an essential component of root response to warming temperatures [77,78]. Among the processes uncovered by our analysis of transcriptomic changes triggered by warm temperatures in *B. napus* roots, we also found an activation of several hormonal responses. We detected induced levels of two ET biosynthesis genes, *BnACO1* and *BnSAM*, and of two ABA sensors, *BnPYL4* and *BnPYL5* in the primary roots in response to warm temperatures. Both hormones increase their level under heat stress and are known to mediate thermotolerance in several crops [79,80,81], suggesting that an activation of biosynthesis or signaling of ET and ABA could also control warming response in *B. napus* roots. An analysis of the role of hormonal pathways on the cellular and developmental responses of roots to warming temperatures would be needed to test this hypothesis.

Together with the differential transcriptional responses, we found common pathways of gene activation between varieties in our analysis. Components of ROS/redox-signaling pathway (*BnPRX71*), lipid signaling (*BnACX2*), and splicing machinery (*BnSR30*), together with physiological responses such as sugar metabolism (*BnSUC7*) and metabolic processes that are considered primary temperature-sensing events, displayed altered expression in *B. napus* roots independently of their specific primary root response [40,77,82]. Thus, this activation of genes related to temperature sensing together with the induction of HS core genes reinforces the idea that a common set of mechanisms for sensing and signaling temperature changes is shared between warming and heat stress in *B. napus* roots. Similarly, a conserved transcriptomic response between warming and higher temperatures has previously been observed in Arabidopsis seedlings at very early times upon temperature stress [83]. Following this initial activation, some varieties seem to decrease the HSR to protect their root growth from the negative effect of a maintained stress response. To attenuate the activation of the HSR, several molecular mechanisms could be involved, including specific transcriptional repression. However, none of the key HS transcriptional repressors known to negatively regulate temperature stress response, primarily the HSFsB family [84], were differentially upregulated in the varieties analyzed in our study. Another possibility is through the fine-tuned control of post-translational modifications of HSFs that change their transport, localization, or turnover, setting positive or negative effects [85]. Alternatively, similar epigenetic changes concerning the histone variant H2A.Z could be mediating this process, as they have been shown to mediate the transcriptomic response to warm temperatures in Arabidopsis, [86]. Further analysis will be necessary to fully uncover the mechanisms that contribute to this attenuation mechanism in *B. napus* roots.

Ambient temperatures are gradually rising as a consequence of global warming, negatively impacting crop productivity. This problem will increase even more in agricultural systems, in which increased temperatures are usually accompanied by complex and concomitant detrimental soil conditions, such as enhanced evapotranspiration and compaction of the soil, changes in nutrient composition and moisture, and soil salinization. Root systems will be required to respond to all these heterogeneous soil environments by producing a combination of root traits that ensure plant survival [87]. For example, the production of shallow roots is an effective strategy for a scarcity of water, but when this is accompanied by poor soil nutrient content, root growth and lateral root branching has to be redirected into deeper regions of the soil where these resources are more abundant [88]. Therefore, improving crop tolerance to warming will necessarily require strategies involving the exploitation of combined morphological, cellular, and genetic pathways underlying beneficial root responses (Figure 7D). Challenging differential root responses to combined environmental stresses will uncover the beneficial traits and possible trade-offs of a specific RSA for enhanced tolerance to warming and other associated stresses. In this context, the association between root trait indexes and the induction of specific gene networks, as well as differences in the dynamics of transcriptional heat-shock response such as the ones uncovered in our study may provide the biotechnological tools needed to exploit root response to these complex environmental changes.

## 4. Materials and Methods

### 4.1. Plant Materials and Growth Conditions

A total of 10 *B. napus* SOSR varieties were used (Appendix A). Seeds were germinated in ¼ MS agar plates for 3 days and transferred to a pouch-and-wick system. The system consisted of growth pouches assembled from two moistened black cardboards (42 × 29.7 cm) and overlaid with 0.5 mm polypropylene black covers. The paper and cardboards were clipped together to each side using PVC spine bars. The growth pouches were set vertically into plastic trays, so that the lowest 5 cm of the blotting paper were submerged in 2 L of nutrient solution (¼ MS) (adapted from [89,90]). Each pouch (experimental unit) containing six seedlings of each variety was randomly placed in a growth chamber, where they were subjected to a constant temperature (21 °C/29 °C), 16 h daylight (150 µmol/m^2^/s), 40% RH, and the same watering regime with nutrient solution (¼ MS) for 7 days [91].

For root length measurements at 15 days, 3-day-old seedlings were transferred to perlite pots (15 cm diameter and depth) and watered with either 100 mL/pot of ¼ MS (plants grown at 21 °C) or 200 mL/pot (plants grown at 29 °C) to maintain constant perlite humidity at both temperatures. Perlite pots were randomly placed in a growth chamber, where they were subjected to a constant temperature (21 °C/29 °C), 16 h daylight, and 40% RH.

### 4.2. Root Trait Analysis

For root trait analysis, three independent biological experiments using pouches containing 6 seedlings for each variety and temperature treatment were used. The seedlings were grown in a single pouch with enough separation between each individual seedling to avoid any overlap of the root systems. Then, pictures of intact *B. napus* roots of the three independent experiments grown for 7 days (when the first and second leaves unfold) at 21 °C or 29 °C were taken using a copy stand and a resolution of 314 ppi. Seedlings were not removed for imaging, but individual pictures of each seedling from the same pouch were taken for imaging. For quantification of root traits, the GiaRoots semi-automated software v0.1 was used [92]. Secondary root number was quantified using saRIA v0.1 [93]. Two-way ANOVA analysis of the contribution of genotype (G), temperature (T), and their interaction (GxT) to the variance of root trait values after warm temperature treatment were performed using GraphPad Prism 6 software. After testing the data for normal distribution using the Shapiro test (significance level 0.05), and for homoscedasticity, using the Brown–Forsythe and Welch test, grouped individual data corresponding to the 3 biological replicates were treated as no matched data for regular (not repeated measures) two-way ANOVA followed by a multiple comparison unpaired t-test with the desired false discovery rate (Q) value set to 1.000%. Traits that did not pass the normality test were compared using a non-parametric Kruskal–Wallis test. Principal component analysis (PCA) was performed on scaled data using the corresponding command from the FactoMiner R package. Hierarchical clustering was performed using the agnes command from the Cluster R package. The optimal number of clusters was determined using K-means cluster analysis and the Elbow method in R. Three independent experiments containing 5 seedlings of each variety were used for quantification of the total root length of plants grown in perlite pots for 15 days at 21°/29 °C. Seedlings were grown individually in each pot, each seedling was removed from the pot, and the roots were washed carefully to remove the perlite. Then, each seedling was placed against a black background with the roots extended, and an individual picture was taken. Total root length was quantified using the Fiji software 1.53k [94].

### 4.3. Cellular Parameter Analysis

Root tips from three independent biological replicates using three experimental units (3 experimental units, 18 seedlings, of each variety and temperature treatment per replicate) of *B. napus* pouch-and-wick-grown seedlings of each variety and temperature treatment (21 °C and 29 °C) were stained using the modified pseudo-Schiff propidium iodide (mPS-Pi) method, as described in [95]. Sequential pictures of the root tip (from meristematic zone up to first differentiated cells) were taken using a vertical Confocal Zeiss LSM-880-Axio-Imager2 microscope with an LD/LCI-Plan-Apochromat 25×/0.8 Imm Korr-DIC-M27 objective (Carl Zeiss Microscopy GmbH, Jena, Germany). The excitation wavelength was 561 nm, and emission was collected at 568–702 nm. Pictures were manually adjusted and stitched using Fiji [94]. Cell number and size of the external cortical layer were quantified using Cell-o-Tape macro [96]. Root meristem size was measured from the QC to the first cell on the cortex that doubled its size at the start of the elongation zone. The apical meristem is defined as the part of the meristem in which the cells are continually dividing and expanding at a continuous rate. Therefore, its size was measured from the QC to the first notably larger cortical cell. Then, the basal meristem was measured from the end of the apical meristem to the elongation zone, where the first cell exceeds more than twice its size.

Cell proliferation analysis was performed by in vivo 5-ethynyl-29-deoxy-uridine (EdU) labeling as described in [97], using root tips from three independent biological replicates (1 experimental unit for each variety and temperature treatment per replicate). EdU is a thymidine analogue in which a terminal alkyne group replaces the methyl group in the 5th position. EdU is incorporated into newly synthesized DNA during DNA replication by cells in the root sample. A fluorescent azide, iFluor-488, is then added that diffuses freely through native tissues and DNA, and it covalently cross-links to the EdU in a “click” chemistry reaction. DAPI (4′,6-diamidino-2-phenylindole) is a fluorescent stain that binds strongly to adenine–thymine-rich regions in DNA. Detection of EdU was performed using the baseClick-Edu488 kit (BCK-EdU488) according to the manufacturer’s instructions (baseclick GmbH, Munich, Germany). Two µg/mL 4′,6-diamidino-2-phenylindol (DAPI) was used for DNA counterstaining. For mitotic index quantification, plants grown for 6 days in the pouch-and-wick system were treated for 2 h by submerging their roots in ¼ MS solution containing 10 µM 5-EdU (pulse). After the pulse period, 5-EdU was rinsed with water, and plants were grown in ¼ MS for 8 h (chase period), before root tips were collected for EdU detection. Sequential image stacks of the meristematic zone were taken using an excitation wavelength of 488 nm and emission collection at 498–598 nm for an EdU or excitation wavelength of 405 nm and emission collection at 410–498 nm for DAPI. Image stacks were projected and stitched using Fiji [94]. EdU- and DAPI-stained nuclei were analyzed using CellProfiler software v3.1.9 [98].

### 4.4. RNA Extraction and Sequencing Analysis

Total RNA from three independent biological replicates (3 experimental units, 18 seedlings, for each variety and temperature treatment per biological replicate) was extracted from root tips of pouch-and-wick-grown *B. napus* seedlings under 21 °C or 29 °C. Root tips of 0.5 cm, corresponding with the distance from the tip up to the first differentiated cell (root hair cell), was dissected, and RNA was extracted using TRIzol reagent (Invitrogen, Boston, MA, USA). Total DNA-free RNA was cleaned using RNeasy Plant Mini Kit purification columns (Qiagen, Hilden, Germany). RNA quality and integrity were assessed on the Agilent 2200-TapeStation (Agilent, Santa Clara, CA, USA). Library preparation was performed using 1μg of high-integrity total RNA (RIN > 8) using the TruSeq Stranded mRNA library preparation kit. Libraries were sequenced on an Illumina-Hiseq2000 platform using paired-end sequencing of 125 bp in length. Quality control of RNA-seq reads was performed using FastQC software v0.11.1. Quality filtered reads were mapped to the rapeseed genome (AST_PRJEB5043_v1 [99]) using HISAT2 [100]. Differential expression analysis of raw count data was performed using DESEQ2 [101] in R. Correlation analysis of differentially expressed genes was performed using ggplot2 in R. Hierarchical clustering of transcriptomic data was performed using Cluster 3.0 [102]. Over-represented biological functions of gene clusters were assessed using SeqEnrich [103].

### 4.5. Expression Analysis by qRT-PCR

Total RNA from three biological replicates (3 experimental units, 18 seedlings, for each variety and temperature per biological replicate) independently collected from replicates used in RNA-seq analysis was extracted from root tips of pouch-and-wick-grown *B. napus* seedlings under 21 °C or 29 °C. First-strand cDNA was synthesized with the RevertAid First Strand cDNA Synthesis Kit (Thermo Scientific, Waltham, MA, USA). qPCR was performed using the LightCycler^®^ 480 SYBRGreen I Master (Roche, Basel, Switzerland) on a LightCycler^®^ 480 System (Roche, Basel, Switzerland). All the expression analysis was performed using three biological and three technical replicates. Expression values were normalized with those of *BnACT7* [104].

### 4.6. Statistical Analysis

Quantification data were analyzed using GraphPad Prism 6 software. All statistical analyses, one-way ANOVA (Tukey’s multiple comparisons test), two-way ANOVA, and *t*-test were performed with built-in analysis tools and parameters.

## Figures and Tables

**Figure 1 ijms-24-01143-f001:**
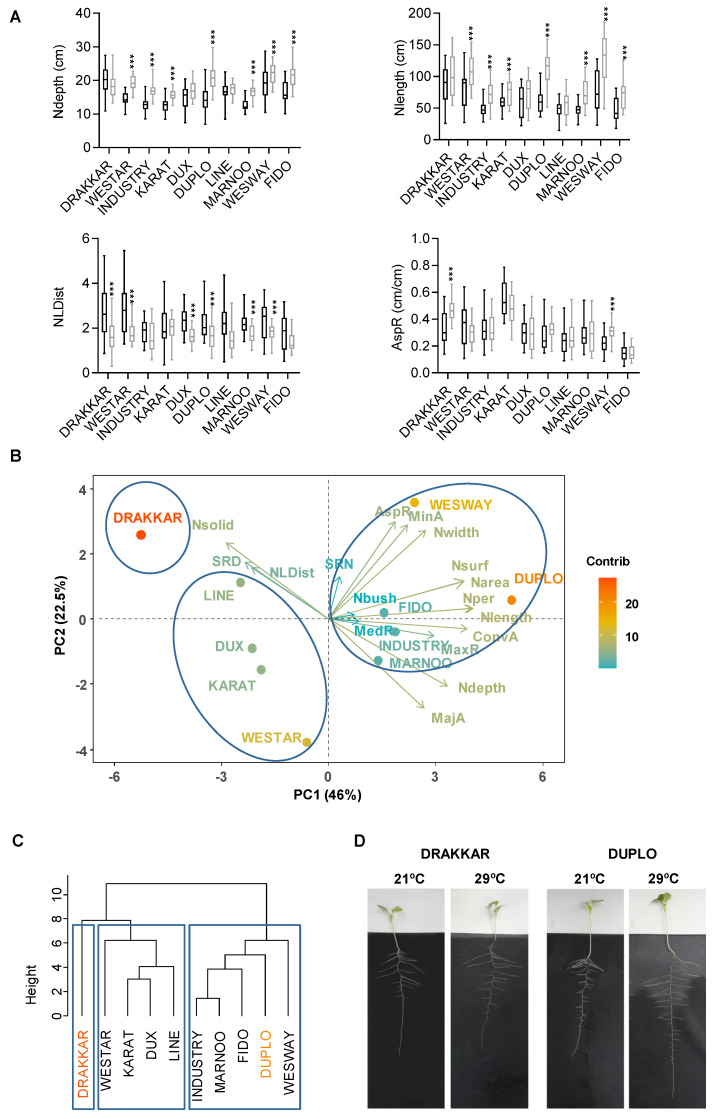
Differential RSA changes in *B. napus* varieties led to an increase in soil exploration in response to warm temperature. (**A**) Differential values of representative root trait categories: extent (network depth (Ndepth, cm)), size (network length, (Nleght, cm)), distribution (network length distribution (NLDist)), and shape (ellipse axis ratio (AspR), cm cm^−1^) traits of a collection of 10 spring oilseed rape genotypes grown at 21 °C and 29 °C. Statistical *t*-test analysis, *** FDR < 0.01. (**B**) Biplot of principal component analysis (PCA) based on all root traits analyzed, showing the high contribution of size and shape traits to variability of SOSR genotypes. Traits and varieties are colored based on contribution to the variance. Circles highlight the three different groups based on their root response to warming. (**C**) Dendrogram plot of SOSR genotypes. AGNES (agglomerative nested) hierarchical clustering was used, where *Y*-axis represents (dis)similarity based on Ward’s minimum variance method. Color boxes highlight three main clusters according to their root trait values. (**D**) Representative root organization of Drakkar and Duplo *Brassica napus* varieties grown in a pouch-and-wick system for 7 days at 21 °C or 29 °C.

**Figure 2 ijms-24-01143-f002:**
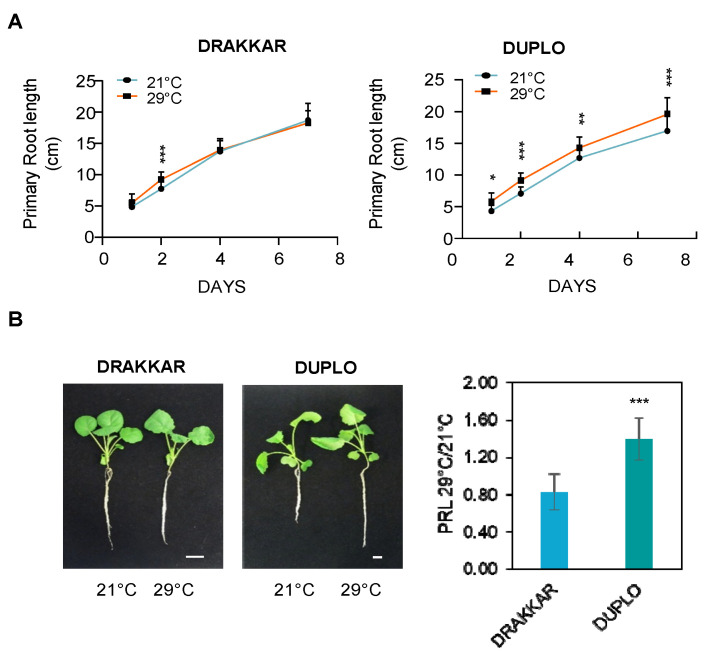
*B. napus* primary root differentially responds to warm temperatures. (**A**) Dynamics of primary root growth in Drakkar and Duplo varieties grown at 21 °C and 29 °C. Graph represents means for 3 independent biological replicates. Statistical *t*-test analysis * *p* < 0.01, ** *p* < 0.001, *** *p* < 0.0001. (**B**) Ratio of root length and representative images of Drakkar and Duplo root phenotypes of 15-day-old seedlings grown at 21 °C and 29 °C in perlite pots. The ratios were similar to the values previously reported at 7 days. Scale bars, 5 cm. Statistical *t*-test analysis *** *p* < 0.0001.

**Figure 3 ijms-24-01143-f003:**
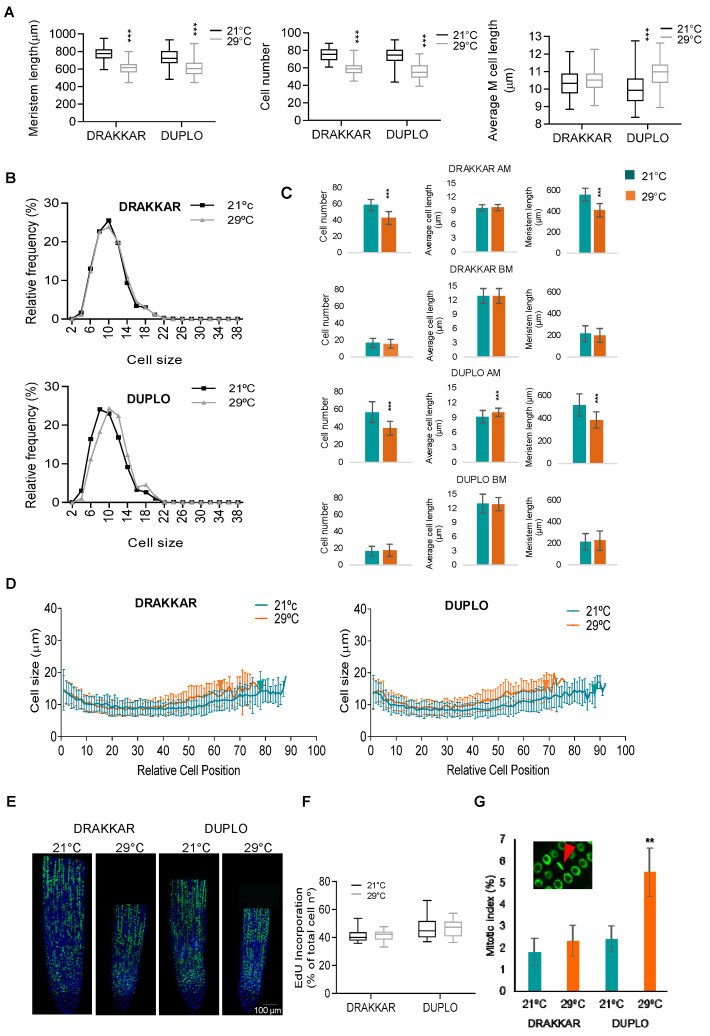
Root meristem changes underlie differential root responses to warm temperatures in *B. napus*. (**A**) Meristematic cell number, meristem length, and average meristematic cell length of roots of Drakkar and Duplo varieties grown at 21 °C and 29 °C. Both varieties presented shorter root meristems at 29 °C than at 21 °C that correlated to a reduced number of meristematic cells in both genotypes. Statistical *t*-test analysis, *** FDR < 0.01. (**B**) Distribution of cell size frequencies of meristematic cells in roots of Drakkar and Duplo varieties grown at 21 °C and 29 °C. Warming resulted in a reduction in the number of shorter cells and a consistent increase in the frequencies of longer-sized cells. (**C**) Meristematic cell number, meristem length, and average meristematic cell length of cells in the apical (AM) and basal (BM) meristems of Drakkar and Duplo varieties grown at 21 °C and 29 °C. Statistical *t*-test analysis, *** *p* < 0.0001. (**D**) Average cell length of the meristematic cells according to their relative position in the root meristem, from the first cell close to the QC, considered as position 0 in the *X*-axis, to all the cells up to the last meristematic cell before the elongation zone, of Drakkar and Duplo varieties grown at 21 °C and 29 °C. The boundary between apical and basal meristem starts closer to the QC when roots are grown at warm temperature. Blue and orange arrows mark the boundary between apical and basal meristems at 21 °C and at 29 °C, respectively. (**E**) Confocal microscope images of Drakkar and Duplo double-labeled 5-ethynyl-29-deoxy-uridine (EdU, green) and 4′,6-diamidino-2-phenylindol (DAPI, blue) root meristems grown at 21 °C and 29 °C. Scale bar, 100μm. (**F**) EdU incorporation ratios (% EdU-labeled nuclei from DAPI-labeled nuclei) in Drakkar and Duplo root meristems grown at 21 °C and 29 °C. Statistical *t*-test analysis found no significant differences. (**G**) Relative number of mitotic cells (mitotic index) as the percentage of EdU-labeled M-phase nuclei out of the total number of EdU-labeled nuclei of Drakkar and Duplo root meristems grown at 21 °C and 29 °C. Statistical t-test analysis, ** *p* < 0.001. Inlet shows EdU-labeled (green) nuclei in M-phase (red arrow).

**Figure 4 ijms-24-01143-f004:**
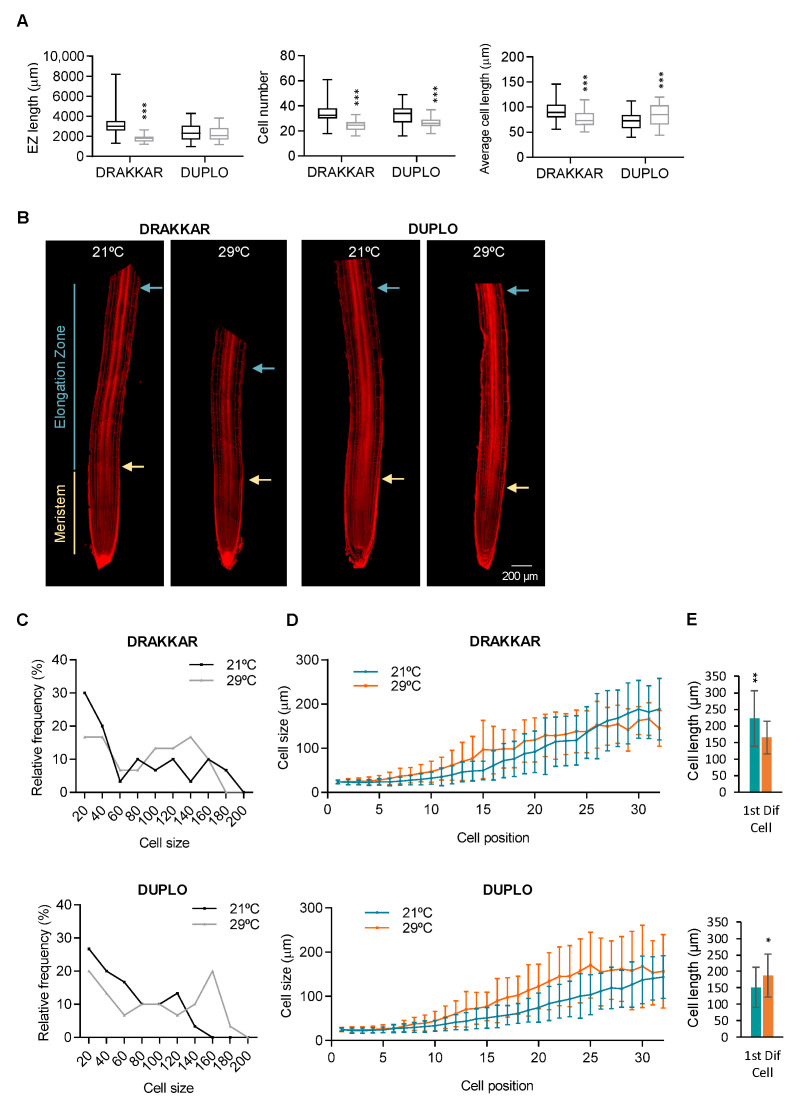
Warm temperatures increase cellular elongation in *B. napus* roots. (**A**) Elongation zone length, cell number, and average cell length of cells in the root EZ of Drakkar and Duplo varieties grown at 21 °C and 29 °C. Statistical *t*-test analysis, *** FDR < 0.01. (**B**) Confocal microscope images of mPS-PI-stained roots of 5-day-old seedlings of Drakkar and Duplo varieties grown at 21 °C compared to 29 °C. Yellow and blue arrows indicate the boundary between the root meristem and the EZ, and the EZ and the differentiation zone, respectively. Scale bars, 200 μm. (**C**) Cell size frequency distribution of the first 30 cells in the root elongation zone of Drakkar and Duplo varieties grown at 21 °C and 29 °C. (**D**) Average cell length of EZ cells according to their relative position in the root of Drakkar and Duplo varieties grown at 21 °C and 29 °C. (**E**) Cell length of the first cell of the differentiation zone in Drakkar and Duplo roots grown at 21 °C (green) and 29 °C (orange) Statistical *t*-test analysis * *p* < 0.01, ** *p* < 0.001.

**Figure 5 ijms-24-01143-f005:**
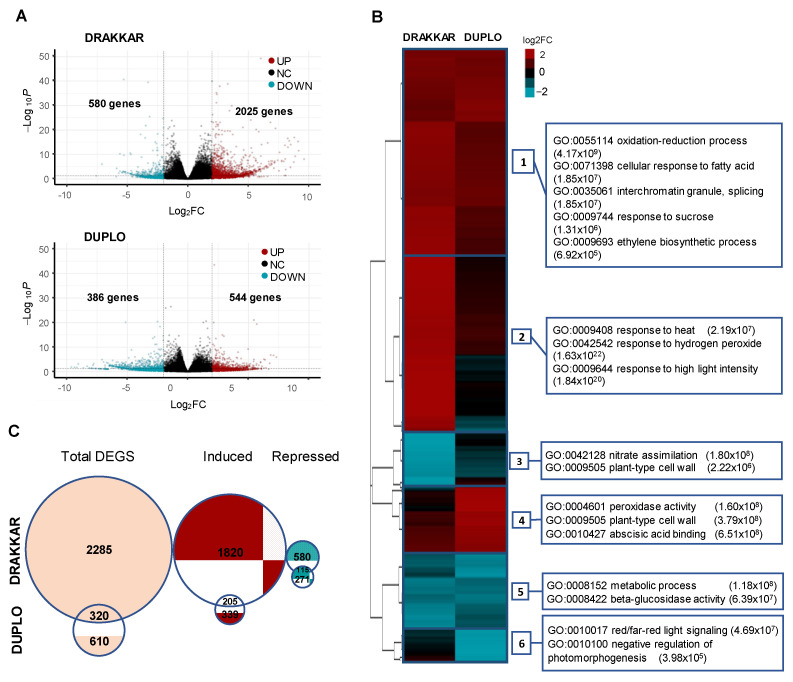
*B. napus* genotypes modify their transcriptional programs to modify their root growth to respond to warm temperatures. (**A**) Volcano plot of average gene expression changes and number of all differentially expressed genes (DEGs) of Drakkar and Duplo root tips grown at 21 °C compared to 29 °C, representing a total of 3215 unique genes with altered expression at 29 °C. *x*-axis represents fold changes (Log2FC), and *y*-axis represents statistical significance (−Log10 of *p* value, −Log10p). The dashed lines show where −1 > log2FC > 1 and adjusted *p* value > 0.05 are located. Maroon dots represent upregulated genes, blue dots are downregulated, and black-grey dots represent no significant gene expression change. (**B**) Transcriptional changes of DEGs in response to warm temperatures in Drakkar compared to Duplo. Differential genes were hierarchically clustered into six groups, based on differences in log2FC ratio. Upregulated genes are represented in maroon, and downregulated genes are in blue. Each cluster of the heatmap is accompanied on the right by the most significant functional categories (GO categories) and their corresponding *p*-value in brackets. (**C**) Venn diagram showing total (orange), upregulated (maroon), and downregulated (blue) DEGs of Drakkar and Duplo root tips grown at 21 °C compared to 29 °C).

**Figure 6 ijms-24-01143-f006:**
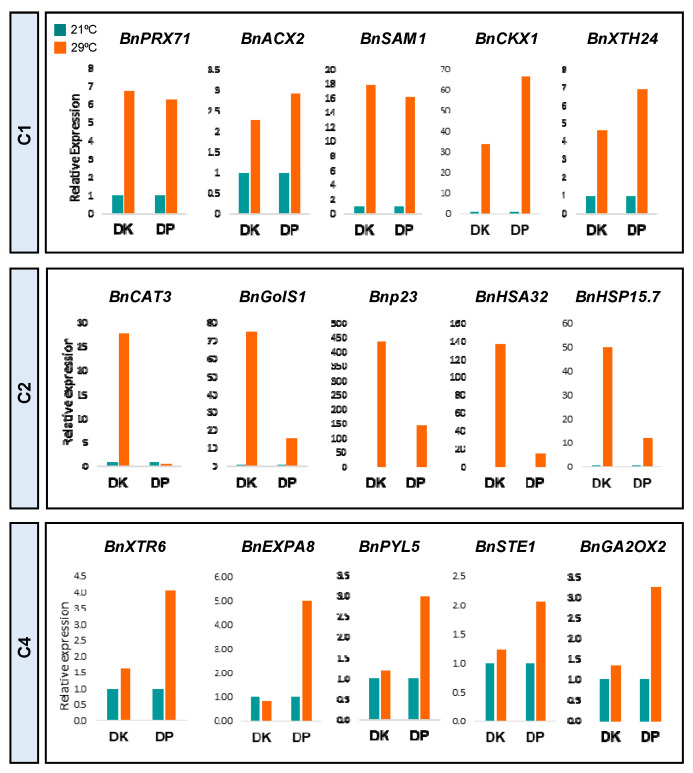
Transcriptional dynamics are altered in response to warm temperatures in *B. napus* roots. Gene expression levels of several genes representative of the main GOs enriched in three of the most significant transcriptional response patterns identified by hierarchical clustering analysis of Drakkar (DK) and Duplo (DP) root tips grown at 21 °C compared to 29 °C. As measured using quantitative RT-PCR (qPCR), relative gene expression values of genes from Cluster 1 were related with the response to oxidative stress (*BnPRX71*), fatty acids (*BnACX2*), cell wall (*BnXTH24*), as well as ethylene biosynthesis (*BnSAM1*) and cytokinin catabolism (*BnCKX1*) and showed similar pattern of expression in both varieties. Meanwhile, relative gene expression values of genes from Cluster 2 were related to response to hydrogen peroxide (*BnCAT3*) and high light intensity (*BnGols1*) together with core heat-shock response genes (HSR) such as chaperones (*BnP23*) and heat-shock proteins (HSPs) (*BnHSA32* and *BnHSP15.7*). Cluster 4 was related with cell wall biogenesis and organization (*BnXTR6* and *BnEXPA8*), as well as hormonal regulation, such as ABA signalling (*BnPYL5*), brassinosteroids (*BnSTE1*), and gibberellin (*BnGA20 × 2*) metabolism. All of these confirmed the differential expression patterns between varieties in response to warming. All the experiments were performed using three biological and three technical replicates. Expression values were normalized with those of *BnACT7* (Chen et al., 2010). The second and third biological replicates are shown in Appendix AC.

**Figure 7 ijms-24-01143-f007:**
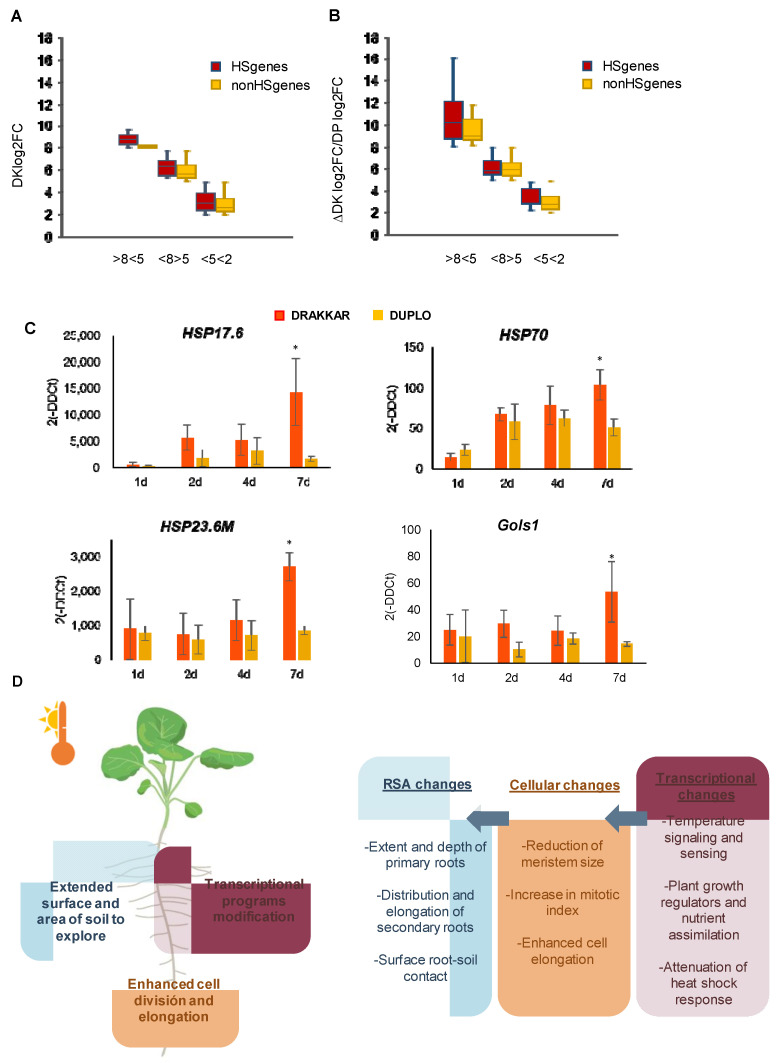
Attenuation of heat-shock stress response is crucial to adjusting root response. (**A**) Comparison between fold change values of heat-shock (HS) response genes and non-heat-shock (nHS) response genes of Drakkar DEGs, showing that this group of HSR genes had the highest fold changes of Drakkar DEGs. *x*-axis represents three consecutive intervals of DK log2FC values (DK, Drakkar). Red boxes correspond to HS genes, whereas yellow boxes correspond to nHS genes. (**B**) Differences between Drakkar log2FC and Duplo log2FC of HS response genes and nHS response genes, showing the highest differences between Drakkar log2FC and Duplo log2FC. *X*-axis represents three consecutive intervals of ∆DK log2FC/DP log2FC values (DP, Duplo). Red boxes correspond to HS genes, whereas yellow boxes correspond to nHS genes. (**C**) Differential dynamics of heat-shock response gene expression (*BnHSP17.6*, *BnHSP70*, *BnHSP23.6M*, and *BnGols1*) at 24 h, 48 h, 4 days, and 7 days after being exposed to 29 °C compared to control temperature, 21 °C, in Drakkar and Duplo roots. Statistical t-test analysis of three biological replicates * *p* < 0.05. (**D**) Warm temperature-triggered changes of several root traits promote root response by facilitating roots’ access to extended areas of soil (blue square box). Enhanced cell division and elongation support this increase in RSA growth (orange square box). Transcriptional changes of different gene regulatory networks related with temperature signaling, plant growth, and nutrient balance drives these cellular changes that result in the differential RSA response (brown square box). Finally, coordinated attenuation of heat response is required for root response to warming temperatures in *B. napus* (brown square box).

**Table 1 ijms-24-01143-t001:** Changes in major root traits of *Brassica napus* varieties result in extended and deeper root systems in response to warm temperatures. Comparative values of mean and coefficient of variation of major root traits classified according to their categories (extent, size, distribution, and shape-related traits) show significant changes in most traits between 21 °C and 29 °C in all *Brassica napus* varieties analyzed. Two-way ANOVA analysis of contribution of genotype (G), temperature (T), and genotype x temperature interaction (GxT) to changes in root trait values after warm temperature treatment were also assessed. Significant differences are indicated by asterisks, * *p* < 0.01, ** *p* < 0.001, *** *p* < 0.0001, and ns shows non-significant differences. Ndepth (network depth, cm), Nwidth (network width, cm), ConvA (network convex area, cm^2^), MajA (major ellipse axis, cm), MinA (minor ellipse axis, cm), Nlength (total network length, cm), Narea (network area, cm^2^), Nper (network perimeter, cm), Nsurf (network surface area, cm^2^), Nbush (network bushiness), NLdist (network length distribution), Nsolid (network solidity, cm cm^−2^), MaxR (maximum number of roots), MedR (median number of roots), SRN (secondary root number), SRD (secondary root density, n cm^−1^), AspR (aspect ratio, cm cm^−1^), Nw/d (network width-to-depth ratio, cm cm^−1^). Detailed descriptions of root traits are provided in Appendix A.

Trait	Category		21 °C				29 °C			Change		ANOVA	
		Mean	s.d	95% CI of the Mean	Coefficient of Variation	Mean	s.d	95% CI of the Mean	Coefficent of Variation		G	T	GxT
**Ndepth** (cm)	Extent	15.43	2.6	[13.57–17.29]	16.85	18.52	2.26	[16.9–20.13]	12.19	↑	***	***	***
**Nwidth** (cm)	Extent	4.91	1.08	[4.14–5.69]	22.05	6.24	1.25	[5.35–7.13]	20.01	↑	***	***	ns
**ConvA** (cm^2^)	Extent	47.23	14.53	[36.84–57.63]	30.77	72.62	18.25	[59.56–85.67]	25.13	↑	***	***	***
**MajA** (cm)	Extent	14.12	2.56	[12.28–15.95]	18.17	16.45	2.2	[14.88–18.02]	13.35	↑	***	***	***
**MinA** (cm)	Extent	4.05	0.90	[3.38–4.71]	22.98	5.08	1.1	[4.3–5.86]	21.59	↑	***	***	ns
**Nlength** (cm)	Size	61.85	15.35	[50.86–72.83]	24.82	87.37	22.97	[70.94103.8]	26.29	↑	***	***	***
**Narea** (cm^2^)	Size	3.15	0.83	[2.56–3.75]	26.4	4.26	1.17	[3.39–5.06]	27.64	↑	***	***	**
**Nper** (cm)	Size	128.5	31.27	[106.1150.8]	24.34	183.1	47.73	[148.9217.2]	26.07	↑	***	***	***
**Nsurf** (cm^2^)	Size	11.46	3.04	[9.28–13.64]	25.55	15.34	4.26	[12.29–18.39]	27.78	↑	***	***	**
**Nbush**	Distribution	3.21	0.55	[2.81–3.61]	17.3	3.36	0.34	[3.11–3.6]	10.23	≈	***	ns	*
**NLdist**	Distribution	2.26	0.33	[2.02–2.5]	14.63	1.68	0.17	[1.56–1.8]	9.88	↓	***	***	*
**Nsolid** (cm^2^ cm^−2^)	Distribution	0.07	0.01	[0.066–0.079]	12.53	0.06	0.01	[0.056–0.064]	10.26	↓	***	***	***
**MaxR**	Distribution	3.31	0.63	[2.86–3.76]	19.14	4.01	0.63	[3.56–4.46]	15.74	↑	***	***	ns
**MedR**	Distribution	1.04	0.05	[1–1.07]	4.56	1.28	0.21	[1.13–1.43]	16.1	↑	***	***	**
**SRN**	Secondary roots	24.68	4. 46	[21.49–27.87]	18.07	34.11	6.48	[29.48–38.75]	18.99	↑	***	***	*
**SRD** (n cm^−1)^	Secondary roots	1.64	0.32	[1.41–1.86]	19.27	1.84	0.21	[1.69–1.99]	11.66	↑	***	***	*
**AspR** (cm cm^−1^)	Shape	0.30	0.10	[0.23–0.38]	34.87	0.32	0.09	[0.25–0.39]	28.92	↑	***	ns	***
**Nw/d** (cm cm^−1^)	Shape	0.32	0.08	[0.26–0.38]	25.28	0.34	0.08	[0.29–0.39]	22.44	↑	***	ns	***

## Data Availability

The RNA-seq data reported in this article have been deposited in the public functional genomics data repository GEO (accession number GSE173272; https://www.ncbi.nlm.nih.gov/geo/query/acc.cgi?acc=GSE173272, accessed on 17 April 2023).

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
