# Peer review of "Brassica napus Roots Use Different Strategies to Respond to Warm Temperatures"

_ijms, 2023, doi:10.3390/ijms24021143_

Round 1

Reviewer 1 Report

I think that figures understanding would improve if the dimension o f the words is enlarged.

It could be an interesting information to know the heritability of the response to the warm temperatures of the different varieties you tested 

Author Response

Response to Reviewers´ Comments for the Author:

Answers to all comments and suggestions are addressed after each point raised by the reviewer (in cursive):

Reviewer 1
Comments and Suggestions for Authors

  1. I think that figures understanding would improve if the dimension of the words is enlarged.

-We have enlarged the dimensions of the words in all the figures. We have increased the font size between two to four times in all the figures as well as increased the size of the graphs and pictures. To allow for these modifications, in Figure 6, we have used abbreviations for the name of the varieties, DK for Drakkar and DP for Duplo, in order to increase graphs‘size in all the panels. We believe that this complete redoing of all the figures in the manuscript, including the Supplemental figures, will substantially improve their understanding as the reviewer suggested.

  1. It could be an interesting information to know the heritability of the response to the warm temperatures of the different varieties you tested.

- We agree with the reviewer that the heritability of the response to warm temperatures of the different varieties could be a valuable information. Unfortunately, the experimental design was not planned to obtain the data needed to analyse this parameter with accuracy. Given the life cycle of the Brassica napus plants and the necessity to replicate measurements for all the traits with parental lines and their progeny, we consider that the calculation of this heritability would be out of the framework of this paper.

Reviewer 2 Report

This paper was a very well-designed results for the root of Brassica napus. I am also doing brassica research and have also studied the roots of legumes. 

It is very important to study the roots of crops. Nevertheless, it is a very difficult area to start researching as it takes a lot of time and effort.

 The methods the authors performed in this experiment may be attractive suggestions for root researchers.

authors should write the results in detail, it will be helpful for readers to understand. I think there is a part omitted.

1. The authors recommend checking the Bartlett test and necessary assumptions before applying PCA. You must have checked it, but I think you omitted it.

2. If the eigenvector is 1 or more, it is considered that there are 2 PCs.

3. The authors clustered them into three groups using hierarchical clustering. 

I don't think the authors arbitrarily divided it into three. For example, I think I have confirmed that the optimal clustering is 3 using k-means. 

It should be mentioned how they are divided into three groups.

4. The author mentioned Bnp23, BnHSA32, and BNhsp15.7 as genes related to heat shock response in fig 6, 

but the genes used to prove the hypothesis were BnHSP17.6, BnHSP70, and BnHSP23.6. Adding an explanation of this will help the reader understand.

5. There are Figures where legends are omitted. 

Author Response

Response to Reviewers´ Comments for the Author:

Answers to all comments and suggestions are addressed after each point raised by the reviewer (in cursive):

Reviewer 2
Comments and Suggestions for Authors

1. The authors recommend checking the Bartlett test and necessary assumptions before applying PCA. You must have checked it, but I think you omitted it.

- We have included in the material and methods section (Subsection 4.2. Root trait analysis) the tests used to process the data before applying the PCA analysis as suggested by the reviewer.

2. If the eigenvector is 1 or more, it is considered that there are 2 PCs.

- We agree with the statement made by reviewer 2. In our case we choose PC1 and PC2, with eigenvalues 8.28 and 4.18, respectively.

  1. The authors clustered them into three groups using hierarchical clustering. I don't think the authors arbitrarily divided it into three. For example, I think I have confirmed that the optimal clustering is 3 using k-means. It should be mentioned how they are divided into three groups.

- As the reviewer suggested we have determined the optimal number of clusters using K-means cluster analysis but we have also used the Elbow method, the average silhouette method and the gap statistic method in R. We have included this information in the material and methods section (Subsection 4.2. Root trait analysis) and added a new sentence in the results section (“Thus, the varieties were grouped in three clusters according to the prevailing combination of traits displayed to respond to warming (Figure 1C). This clustering underlies the same phenotypic categorization of the varieties obtained in the previous analysis”) to clarify how they are divided into three groups as the reviewer suggested.

  1. The author mentioned Bnp23, BnHSA32, and BNhsp15.7 as genes related to heat shock response in fig 6, but the genes used to prove the hypothesis were BnHSP17.6, BnHSP70, and BnHSP23.6. Adding an explanation of this will help the reader understand.

-We have included the following sentence and two references (Dong et al, 2015; Huang et al, 2022) [42-43]) (marked below in cursive and between quotation marks) to explain the election of these group of genes for further analysis as the reviewer suggested (“To test this hypothesis, we analysed the transcriptional dynamics of several HSR genes belonging to Cluster 2 at 24h, 48h, 4 days and 7 days after being exposed to control temperature, 21ºC, and 29ºC in both varieties. Thus, we monitored the pattern of expression of “four well-known core HSR genes, including the small heat-shock protein, BnHSP17.6; the ATP-dependent chaperone of HSP70 family, BnHSP70; the mitochondrion-localized small heat shock protein, BnHSP23.6M and the HS–induced galactinol synthase, BnGolS1 by qPCR (Figure 7C)[42-43]”. As expected, we found that all these genes started to increase their expression...”).

-New references:

-[42] Dong X, Yi H, Lee J, Nou IS, Han CT, et al. Global Gene-Expression Analysis to Identify Differentially Expressed Genes Critical for the Heat Stress Response in Brassica rapa. PloS One 2015. 10(6): e0130451.

-[43] Huang, L.-Z.; Zhou, M.; Ding, Y.-F.; Zhu, C. Gene Networks Involved in Plant Heat Stress Response and Tolerance. Int. J. Mol. Sci. 2022, 23, 11970.

  1. There are Figures where legends are omitted.

- We have moved the legend of Table 1 from above the Table 1 to below in order to better clarify which figure it belongs to. We have highlighted the legends of the Supplemental Figures that were included in the Supplemental Materials section of the manuscript and we have additionally included them into the Supplemental Figures.

Round 2

Reviewer 2 Report

I am grateful to the authors for carefully reflecting my comments.